# Knowledge of Predatory Practices within the Substance Use Disorder Treatment Industry: Development of a Measurement Instrument

**DOI:** 10.3390/ijerph19137980

**Published:** 2022-06-29

**Authors:** Antoinette Y. Farmer, Yuhan Wei, Kristen Gilmore Powell, Peter Treitler, Amal Killawi, David Lardier, N. Andrew Peterson, Suzanne Borys, Donald K. Hallcom

**Affiliations:** 1School of Social Work, Rutgers, The State University of New Jersey, New Brunswick, NJ 08901, USA; yw623@ssw.rutgers.edu (Y.W.); krisgil@ssw.rutgers.edu (K.G.P.); peter.treitler@rutgers.edu (P.T.); ak1394@ssw.rutgers.edu (A.K.); andrew.peterson@ssw.rutgers.edu (N.A.P.); 2Center for Prevention Science, Rutgers, The State University of New Jersey, New Brunswick, NJ 08901, USA; 3Department of Psychiatry and Behavioral Sciences, University of New Mexico School of Medicine, Albuquerque, NM 87131, USA; dalardier@salud.unm.edu; 4Division of Mental Health and Addiction Services, New Jersey Department of Human Services, Hamilton, NJ 08691, USA; suzanne.borys@dhs.nj.gov (S.B.); donald.hallcom@dhs.nj.gov (D.K.H.)

**Keywords:** knowledge, predatory practices, scale development, loved ones, substance use disorder, substance use disorder treatment

## Abstract

The increase in predatory practices in the substance use disorder treatment industry calls for the development of measures to assess individuals’ knowledge about these practices. Methods: This study describes the development of the Knowledge of Predatory Practices Scale (KPPS), a newly developed measure designed to assess the knowledge of predatory practices within the substance use disorder treatment industry. An exploratory factor analysis was conducted to determine the factor structure of this measure. Results: The final 11-item KPPS consisted of two factors—knowledge about general predatory practices (9 items) and knowledge about unethical practices (2 items). Overall, these factors explained 61.75% of the total variance. The Cronbach’s alpha for the KPPS was 0.81. Conclusions: The KPPS is a reliable measure of knowledge of predatory practices within the substance use disorder treatment industry and can be used as a measurement tool to educate individuals seeking help for their loved ones who are misusing substances.

## 1. Introduction

Substance use disorder (SUD) in the United States has been deemed an epidemic [1]. Approximately, 100,306 drug overdose deaths were reported in the United States at the end of April 2021 for a 12-month period according to the Centers for Disease Control [2]. This number is higher than that reported in 2020 for the same time period. Growing rates of substance use have also had a profound economic impact. For example, in 2019, the economic cost of the opioid crisis to the United States was estimated to be USD 631 billion, including health care costs, productivity loss, substance-misuse premature death, childcare/family assistance, and criminal justice involvement [3,4].

Substance use has not only accelerated a public health crisis but also generated an increase in health care fraud. In 2020, the Department of Justice (DOJ) brought charges against criminal defendants for opioid-related submissions of USD 6 billion in false and fraudulent claims to Medicare, Medicaid, TRICARE, and private insurance companies for treatment, including USD 845 million for substance abuse treatment and more than USD 30 million for illegal opioid distribution [5]. The submission of false and fraudulent claims is the result of unethical treatment practices, including patient brokering, unnecessary services, and overcharging. Patient brokering—unlicensed individuals or facilities finding individuals misusing substances and referring them to specific treatment facilities in exchange for money or other benefits—is a common unethical practice in many states [6]. In many cases, patients are referred to facilities regardless of whether it is the best fit for their needs. Treatment providers also conduct other types of predatory practices, such as manipulating online search results of treatment provided and their effectiveness, charging excessive fees, and charging for unnecessary services [7]. Family members of individuals misusing substances looking for treatment for their loved ones often use Google or other search engines to find treatment, becoming prey to unscrupulous treatment providers due to manipulated search results [7]. While in treatment, clients’ insurance is billed for unnecessary treatment or overpriced services [7,8]. Clients may be charged excessive fees for drug tests, sober living, and out-of-network services despite being treated in in-network facilities [7,8]. Clients are also targeted for out-of-state treatment, resulting in out-of-pocket expenses [7,8].

With the increase in predatory practices, some state governors and senators have introduced or signed legislation to prohibit unethical practices, but further regulation is needed. For example, in 2018, former Governor Cuomo of New York signed legislation to prohibit New York SUD treatment programs from using “patient brokers” [9]. Meanwhile, in February 2021, California Senator Thomas Umberg introduced “The California Ethical Treatment of Persons with Addiction Act” (Senate Bill 349) [10] to protect individuals seeking treatment for SUD from predatory practices and called for creating tools to root out, punish, and deter unethical practices such as patient brokering, over charging for drug testing, and recovery residence. In March 2021, New Jersey passed a bill (A-2280) to deter SUD treatment centers from patient brokering. Although some states have put into place legislation to address predatory practices, there is a considerable number of states where such legislation has not been passed [3].

Given the increase in predatory practices related to SUD treatment, Peterson et al. [11] developed the Knowledge of Predatory Practices Scale (KPPS), the first measure to assess knowledge of predatory practices related to SUD treatment. Until now, little research has been conducted to determine how prevalent predatory practices are in the SUD treatment industry or how aware or knowledgeable individuals misusing substances, or their family members are about these practices. To our knowledge, this is the first study to examine the knowledge of predatory practices of family members of individuals misusing substances and the first study to examine the psychometric properties of a measure developed to assess such knowledge.

The KPPS was developed by reviewing the National Association of Addiction Treatment Providers (NAATP) Enhanced Ethics Compliance and Consumer Protection Initiative document. The NAATP published the Enhanced Ethics Compliance and Consumer Protection Initiative in 2017, which provided guidance for identifying unethical practices [8]. Unethical practices discussed in the NAATP report include patient brokering, predatory web practices, urinalysis abuse, up-charging and overutilization, disguised “treatment” billing, bait and switch-out-of-network schemes, kickbacks, clinical misrepresentations, and paid call center/directory/call aggregation [8].

The KPPS was developed in the context of an evaluation of Family Support Centers (FSCs) in New Jersey, a program continuing to be funded by the State’s Opioid Response grant. FSC services include offering educational resources to family members seeking information on SUD treatment, educating family members about naloxone training, linking family members to support resources, and educating family members about navigating the treatment system and detecting unethical practices [12]. The KPPS is meant to be used to assess the knowledge of family members and other loved ones about navigating the treatment system and detecting unethical practices of treatment providers targeting individuals seeking SUD treatment. The purpose of this study was to examine the factor structure of the KPPS using an exploratory factor analysis. Additionally, we sought to assess the internal reliability of this measure.

## 2. Materials and Methods

### 2.1. Sample and Procedures

A secondary analysis of the baseline data collected by the Family Support Centers (FSC) evaluation was conducted. The sample included 173 participants who attended FSCs in New Jersey. Prior to receiving supportive services from FSCs, participants completed a self-administered questionnaire assessing their knowledge of predatory practices related to SUD treatment. They also provided demographic and other pertinent information related to their loved ones’ drug usage and treatment history. Before administering the questionnaire, institutional review board approval was obtained for the data collection procedures involving human subjects. Signed consent forms were completed by everyone who participated in the study. Descriptive statistics for the sample are presented in Table 1.

### 2.2. Measures

*Knowledge of Predatory Practices Scale (KPPS).* To measure family members’ knowledge of predatory practices related to SUD treatment, the KPPS [11] was used. The scale was developed to assess family members’ awareness of these practices at enrollment and whether any changes occur throughout their participation in the FSC.

The research team developed an initial item pool based on the review of the NAATP initiative document and clinical experiences of the team members. Once the item pool was developed, the items were sent to experts in the SUD treatment field for their review and feedback. Based on the feedback, the KPPS was finalized, consisting of 13 items. Example items included the following: “Patient brokering is when a provider gives out money or perks in exchange for a patient or a potential patient” and “Some programs claim to have very high success rates even when those rates are determined in an unreliable way”. Items were measured using a 4-point confidence-weighted true/false format, with the options of “I am sure the statement is true/false” and “I think the statement is true/false, but I am unsure.” The confidence-weighted true/false (CFT) format not only estimated the number of correct answers, but also provided the confidence level of participants on the knowledge tested. In this way, the confidence-weighted response format could better evaluate the level of knowledge [13].

According to the confidence-weighted true/false response format score-assigning rule, each response option was assigned a score from −2 to +2. For example, for item 1, choosing “I am sure the statement is true” was assigned a score of +2; choosing “I think the statement is true, but I am unsure” was assigned a score of +1; choosing “I think the statement is false, but I am unsure” was assigned a score of −1; and choosing “I am sure the statement is false” was assigned a score of −2. Answers to all items on the scale, except for items 3, 4, and 13, were assigned +2 for choosing “I am sure the statement is true”. Because items 3, 4, and 13 were negatively worded, choosing “I am sure the statement is false” was assigned +2, and choosing “I am sure the statement is true” was assigned −2. We reverse-coded items 3, 4, and 13 for analysis purposes. Therefore, a higher score represented a higher correctness and confidence level on the KPPS. The mean of each item was calculated, and the percent of correct responses for each item was also calculated. The percent of correctness represented the proportion of participants choosing correct answers with the most confidence, namely, participants who scored +2 on each question.

*Demographics.* Participants were asked to identify their gender as male, female, transgender, or non-binary. Because no respondents identified as transgender or non-binary, the gender variable was recoded as female, where female = 1 and male = 0, with male as the reference group.

*Relationship Status.* The relationship of the family member to their loved ones was assessed by a single question asking the participants to identify their relation to their loved ones. Responses to this question were: 1 = parent, 2 = child, 3 = sibling, 4 = grandparent, 5 = spouse, 6 = significant other, and 7 = other.

*Race/ethnicity*. Participants were asked to respond to two questions inquiring about their race/ethnicity. First, they were asked to identify their ethnicity, by responding to the question “Are you of Hispanic origin?”, where 1 = “no”, 2 = “yes”, and 3 = “prefer not to answer”. This question was followed by asking the participant to identify their racial background, where 1 = “Alaskan Native”, 2 = “American Indian”, 3 = “Native Hawaiian or Other Pacific Islander”, 4 = “White”, 5 = “Asian”, 6 = “Black or African American”, and 7 = “Other”. Combining the responses from the ethnicity and race questions, we found that only four combined categories were chosen by participants, and no one selected “non-Hispanic/Hispanic Alaskan Native”, “non-Hispanic/Hispanic American Indian”, “non-Hispanic/Hispanic Pacific Islander”, “Hispanic Asian”, or “Hispanic Black”. Therefore, we only had four categories. Additionally, “prefer not to answer” and “other categories” were coded as missing on the combined variable. Therefore, the new variable race/ethnicity included four categories: 1 = “Hispanic White”, 2 = “non-Hispanic Asian”, 3 = “non-Hispanic Black”, and 4 = “non-Hispanic White”.

*Age.* Participants’ age and their loved one’s age were measured as continuous variables in this study by asking participants to write down their age and their loved ones’ age.

*Treatment History.* Participants were asked to indicate if their loved one had received treatment and to check all that applied. The responses were: 1 = “currently in treatment”; 2 = “previously in treatment, completed”; 3 = “previously in treatment, not completed”; 4 = “never in treatment”; 5 = “prefer not to answer”. We created a dichotomous variable—treatment history with “currently in treatment”, “previously been in treatment and completed”, and “previously been in treatment but not completed” coded as 1, and “never in treatment” coded as 0, with “never in treatment” as the reference group. Prefer not to answer was coded as missing.

*Loved Ones’ Primary and Secondary Drug of Choice.* Participants reported their loved one’s primary and secondary drug of choice.

*Opioid Overdose History.* Participants reported whether their loved one had ever overdosed from an opioid. The responses were 1 = “no”, 2 = “yes”, and 3 = “prefer not to answer”. “Prefer not to answer” was coded as missing.

### 2.3. Analytic Strategy

All analyses were conducted using SPSS 23 [14]. Descriptive statistics were used to calculate correct/incorrect answers for each item. The mean score, skewness, and kurtosis were also calculated for each item. An exploratory factor analysis (EFA) was conducted to determine the factor structure of the KPPS and examine the response pattern for each item. There were no missing data for the KPPS; however, there were some missing demographic and other pertinent data. The percentage of missing data for these variables is reported in Table 1.

## 3. Results

### 3.1. Descriptive Results

The average age of the participants was 53.20 (SD = 13.90) years old, and the average age of participants’ loved ones was 31.44 (SD = 8.87) years old. The most common primary drug of choice for the participants’ loved one was heroin, while the most common secondary drug of choice was cocaine. About 41.00% of the participants’ loved ones had experienced an overdose from an opioid. Respondents were not asked about the specific opioid their loved one was using that resulted in the overdose. Other pertinent information about the sample and their loved ones is presented in Table 1.

Table 2 presents the percent of correct responses, mean, skewness, and kurtosis for each item. As can be seen in Table 2, the percent of correct answers for the items ranged from 5.80% to 54.30%, with Item 4 having the lowest percentage of participants choosing the correct response. This item asked participants how common it is for providers to manipulate web search results to hide relationships between referral sources and providers that are owned by the same parent company, and the correct answer is, “I am sure the statement is false”. The mean score of KPPS was 0.84 (SD = 0.56), indicating that the participants were not very confident in their knowledge about predatory practices related to SUD treatment. Skewness and kurtosis patterns for each item and the whole scale were also examined. The skewness (−0.72) and the kurtosis (0.52) values of the entire scale indicated that the distribution of the total scale score was normally distributed. Meanwhile, the skewness for each item was within the range from −2 to 2, which indicated that the distribution of scores was normally distributed. In terms of the kurtosis, we can see that the values for Item 6 (“*Some out-of-state programs try to lure patients to their programs by over-promising what they can offer*”) and Item 7 (“*Up-charging or overutilization is when treatment providers perform unnecessary or excessive services to bill patients at a higher rate*”) were greater than 2, which means we had a leptokurtic distribution for these items [15]. Therefore, these two items had a more peaked distribution as opposed to a normal curve distribution.

### 3.2. Exploratory Factor Analysis Results

Because the KPPS is a newly developed measure, we conducted an exploratory factor analysis (EFA). We used Bartlett’s test of sphericity and the Kaiser–Meyer–Oklin (KMO) measure of sampling adequacy to verify the appropriateness of using an EFA. The results indicated a significant Bartlett’s test of sphericity, X^2^ (78) = 920.22, *p* < 0.001, and a KMO value of 0.87, suggesting that an EFA was appropriate for analyzing our data. Since we had two items that had a leptokurtic distribution in our data, we chose Principal Axis Analysis for factor extraction, which is suitable for non-normally distributed data [16]. Because we believed that the factors derived from the analysis were correlated, we chose to use oblimin rotation. We chose to adopt the convention that items would be retained based on the following: (1) items that had a factor loading higher than 0.40 and (2) items that had cross-loadings with less than a 0.15 difference from the items’ highest factor loadings as failing to meet our criteria for item retention [16]. For factor retention, we used an eigenvalue above 1.

Prior to conducting the EFA, the Mahalanobis Distance approach was used to detect any outliers [15]. The results of this analysis revealed 15 cases with outliers (*p* < 0.001). Therefore, we excluded those cases, resulting in an analytic sample of 158 cases for the EFA.

Table 3 presents the results of the EFA. The EFA indicated that KPPS had two retained factors and 12 out of the 13 items were retained. The results also reveal that Item 4 needed to be deleted, because it negatively loaded on the factor. Therefore, the KPPS was comprised of 11 items. Items 2, 5, 6, 7, 8, 9, 10, 11, and 12 loaded on Factor 1—knowledge about general predatory practices—and all factor loadings were higher than 0.60. Items 3 and 13 were negative-worded items loaded on Factor 2—knowledge about unethical practices—and factor loadings were higher than 0.50. The communalities of the 11 items were all higher than 0.20, and the two factors explained 61.75% of the total variance. To assess the reliability of this measure, the Cronbach’s alpha was computed. The results of this analysis revealed that this 11-item measure had a reliability of 0.81.

## 4. Discussion

The purpose of this study was to examine the factor structure of the KPPS, a newly developed measure designed to assess knowledge of predatory practices related to SUD treatment. The results of the EFA revealed that this 11-item measure is a two-factor scale with acceptable reliability. Factor 1—knowledge of general predatory practices and Factor 2—knowledge about unethical practices. Of the original 13 items, one had to be deleted due to a factor loading lower than 0.40 (Item 1), and the other had to be deleted because it negatively loaded on its factor (Item 4). The deletion of these items does not indicate that they are not good items to be used to assess predatory practices, but it may be that they measure other aspects of predatory practices not currently captured by this measure. In reviewing the deleted items, we noticed that Item 1 was double-barreled. Because of this, participants may have found it difficult to determine if the item was correct or not. It is recommended that this item be written as two separate items. Regarding Item 4, it seems to be worded in such a manner that one would have to know what a term (i.e., parent company) meant, in order to determine if the item was correct or not.

As we mentioned earlier, 15 cases were eliminated from the analysis due to outliers. After reviewing the answering patterns of these participants, we discovered that they were different from the rest of the participants. This difference may be due to how the 15 individuals understood what the items were stating. This finding suggests that more intensive pilot testing of these items is warranted.

Several limitations of this study must be acknowledged. A limitation of this study is the small sample size. We were only able to collect data from 173 family members, and we deleted 15 cases due to outliers. Hence, the resulting analytic sample was composed of 158 respondents. Previous studies using family members as respondents have also had small sample sizes [17]. One possible reason for this is that treatment providers tend to find it challenging to connect with families and keep them engaged throughout the entire intervention period [18]. Although as sample sizes diminish, the impact on factor loadings increases. Despite this, there is agreement that fewer data with the absence of or minimal cross-loadings can be used to identify relatively accurate factor loadings [19,20].

The lack of demographic variability in the sample limited our ability to assess for differences between males and females on their knowledge of these predatory practices. Because the sample was predominately non-Hispanic White, we were unable to assess for knowledge differences across racial/ethnic groups. Due to the cross-sectional nature of the data, we could not assess changes in the participants’ knowledge of predatory practices as a result of their participation in the FSC program. Another limitation of this study is that we were not able to assess the construct or divergent validity of this measure. The results of this study are only generalizable to those who participated in this study. The focus of this study was on assessing the factor structure on this measure; therefore, there was no attempt to assess the predictive validity of this measure. Despite these limitations, this is the first study to our knowledge conducted examining the factor structure a measure developed to assess one’s knowledge of predatory practices. Hence, this study fills an important gap in the literature.

## 5. Conclusions

Based on the results of this study, the 11-item KPPS can be used to gather data from family members to assess their initial knowledge of predatory practices related to SUD treatment, as well as their knowledge after they have completed an educational program aiming to increase their knowledge about predatory practices.

Our study points to some future research directions. Given that some of the items were double-barreled or used a phrase in a context that some of the participants may not have been familiar with, we suggest that these items need to be modified prior to this measure being used in future studies. For example, Item 1 could be revised so that it presents as two items. One item would be “Patient brokering is when a provider gives out money in exchange for a patient or potential patient”, while the other would be “Patient brokering is when a provider gives out perks in exchange for a patient or potential patient”. Regarding Item 4, we suggest that it be rewritten as follows: “It is common for providers to manipulate website results to hide their relationship between referral sources and themselves”. New items may also be developed and better worded to replace the items that were dropped. Researchers may also want to consider generating additional items by asking experts in the field of substance use and using cognitive interviewing and other strategies appropriate for item development.

Future studies could explore whether sources of information about predatory practices affect one’s knowledge level. Further, future research could also examine the psychometrics of the KPPS using different samples, for example, individuals misusing substances, social workers providing treatment to individuals misusing substances, or social workers who are training to provide services to individuals misusing substances and their families. Data from such samples could be used to assess the factor structure of the KPPS across these groups, allowing one to assess the conceptual equivalence of this measure.

Moreover, researchers could assess the construct, divergent, and predictive validity of this measure and attempt to obtain a more heterogeneous sample, so that analyses can be conducted to assess the differences in knowledge of predatory practices between different genders and racial/ethnic groups. Finally, an item-response analysis could be conducted to assess how each item functions, as a way of continuing to assess the psychometric properties of this measure. More specifically, a generalized partial credit model item-response analysis should be used, which is appropriate when individuals are given partial credit for getting some aspect of the item correct.

Future research could also explore the utilization of the KPPS in clinical settings and community-based agencies, as this measure was developed in the context of services being provided in FCSs. Data derived from individuals in these settings could provide us with insight into the level of knowledge of individuals that may not be seeking services for their loved ones who are not misusing substances. Such information would also be valuable to policy makers.

## Figures and Tables

**Table 1 ijerph-19-07980-t001:** Descriptive analyses of demographics.

Variables	Type	Mean	SD	Frequency	Percentage	N
**Gender**	dichotomous					173
-female				149	86.10%	
-male				15	8.70%	
-missing				9	5.20%	
**Relationship Status**	categorical					173
-parent				106	61.30%	
-other				16	9.20%	
-sibling				11	6.40%	
-significant other				11	6.40%	
-spouse				10	5.80%	
-child				8	4.60%	
-grandparent				7	4.00%	
-missing				4	2.30%	
*** Treatment History**	dichotomous			144	83.20%	173
-no treatment				21	12.10%	
-missing				8	4.60%	
**Race/Ethnicity**	categorical					173
-non-Hispanic White				136	78.60%	
-Hispanic White				11	6.40%	
-non-Hispanic Black				5	2.90%	
-non-Hispanic Asian				1	0.60%	
-missing				20	11.60%	
**Age**	continuous	53.20	13.90			173
-missing						13
**Loved one’s age**	continuous	31.44	8.87			173
-missing						13
**Opioid Overdosed History ****	dichotomous					173
-no				73	42.20%	
-yes				71	41.00%	
-missing				29	16.80%	

* Treatment history coded as yes includes “currently in treatment”, “previously been in treatment and completed treatment”, and “previously been in treatment but not completed treatment”. ** Opioid overdose history means history of overdose from an opioid. The respondents were not asked about a specific opioid their loved one was using that resulted in the overdose.

**Table 2 ijerph-19-07980-t002:** Descriptive analysis of items on KPPS.

Item	Item Wording	Mean	SD	Skewness	Kurtosis	Correctness
1	Patient brokering is when a provider gives out money or perks in exchange for a patient or a potential patient.	0.69	1.33	−0.87	−0.56	31.20%
2	Body brokers are people who work to direct patients into out-of-state programs that provide little to no treatment.	0.53	1.28	−0.49	−1.17	25.40%
3	It is ethical for programs or sober homes to pay a third party to send patients to their facility.	0.35	1.47	−0.36	−1.42	28.30%
4	It is common for providers to manipulate web search results to hide relationships between referral sources and providers that are owned by the same parent company.	−0.62	1.25	0.69	−0.82	5.80%
5	Online search results for providers can be misleading because some providers manipulate searches to ensure their programs are among the top results.	1.19	1.06	−1.44	1.23	49.10%
6	Some out-of-state programs try to lure patients to their programs by over-promising what they can offer.	1.36	0.91	−1.87	3.52	54.30%
7	Up-charging or overutilization is when treatment providers perform unnecessary or excessive services to bill patients at a higher rate.	1.37	0.89	−1.9	3.83	53.80%
8	Providers listed as “in-network” with insurance companies may not accept specific plans, resulting in higher-than-expected treatment charges.	1.09	1.14	−1.34	0.82	45.10%
9	Treatment providers can give families descriptions of services that may not be exactly accurate.	1.03	1.19	−1.18	0.21	45.10%
10	The “heads and beds” practice happens when clients are prematurely or rapidly moved through phases of treatment based on the need to maintain a certain number of clients in each phase, rather than clinical readiness.	1.13	1.09	−1.42	1.21	45.10%
11	Some programs claim to have very high success rates even when those rates are determined in an unreliable way.	1.18	1.04	−1.45	1.38	46.80%
12	Some treatment referral call centers that appear independent only refer to facilities owned by their affiliates, regardless of whether the facilities are a good fit for the patient.	1.03	1.1	−1.17	0.37	39.90%
13	It is ethical for treatment providers to have staff who are not clinically trained provide free consultations to assess a potential patient’s appropriate level of care.	0.59	1.52	−0.63	−1.21	39.90%

Note: Percent correct represents the percentage of participants responding with answers that were assigned score a of 2. The mean of Item 4 is negative because most participants answered the question wrong.

**Table 3 ijerph-19-07980-t003:** Rotated factor loadings of KPPS.

Item		Factor Loading	Factor Loading
2	Body brokers are people who work to direct patients into out-of-state programs that provide little to no treatment.	**0.60**	0.03
3	It is ethical for programs or sober homes to pay a third party to send patients to their facility.	−0.05	**0.65**
5	Online search results for providers can be misleading because some providers manipulate searches to ensure their programs are among the top results.	**0.69**	−0.10
6	Some out-of-state programs try to lure patients to their programs by over-promising what they can offer.	**0.80**	−0.07
7	Up-charging or overutilization is when treatment providers perform unnecessary or excessive services to bill patients at a higher rate.	**0.71**	−0.11
8	Providers listed as “in-network” with insurance companies may not accept specific plans, resulting in higher-than-expected treatment charges.	**0.69**	−0.08
9	Treatment providers can give families descriptions of services that may not be exactly accurate.	**0.74**	−0.05
10	The “heads and beds” practice happens when clients are prematurely or rapidly moved through phases of treatment based on the need to maintain a certain number of clients in each phase, rather than clinical readiness.	**0.80**	0.13
11	Some programs claim to have very high success rates even when those rates are determined in an unreliable way.	**0.83**	−0.01
12	Some treatment referral call centers that appear independent only refer to facilities owned by their affiliates, regardless of whether the facilities are a good fit for the patient.	**0.80**	−0.02
13	It is ethical for treatment providers to have staff who are not clinically trained provide free consultations to assess a potential patient’s appropriate level of care.	−0.01	**0.51**

Note: The boldface items in the table are those that met our criteria for item retention (i.e., values greater than 0.40).

## Data Availability

Data available on request due to privacy/ethical restrictions.

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
