# Peer review of "Knowledge of Predatory Practices within the Substance Use Disorder Treatment Industry: Development of a Measurement Instrument"

_ijerph, 2022, doi:10.3390/ijerph19137980_

Round 1
Reviewer 1 Report
The purpose of this study was 96 to examine the factor structure of the KPPS, using an exploratory factor analysis. Additionally, authors sought to assess the internal reliability of this measure.
I found the study well designed and well written and no major concerns are not needed.
The most common primary drug of choice for the participants’ one was heroin, while the most common secondary drug of choice was cocaine. About 200 41.0% of the participants’ loved ones had experienced an overdose from an opioid. This last sentence is not clear. What kind of opioides were evaluated?
I also suggest that the conclusions section should also include real major conclusions. It is only focused on future perspectives for research.
Author Response
Comment 1: The most common primary drug of choice for the participants’ one was heroin, while the most common secondary drug of choice was cocaine. About 200 41.0% of the participants’ loved ones had experienced an overdose from an opioid. This last sentence is not clear. What kind of opioids were evaluated?
Response: The question that asked about the history of opioid overdoses was as follows: Participants reported whether their loved one ever overdosed from an opioid. Participants were not asked about the specific opioid their loved one was using that resulted in the overdose. Therefore, we do know what specific opioid resulted in the overdose. We thank-you for pointing out that we need to indicate that we do not know what specific opioid resulted in the overdose. Therefore, we are now saying the following: About 41.0% of the participants’ loved ones had experienced an overdose from an opioid. Respondents were not asked about the specific opioid their loved one was using that resulted in the overdose.
Comment 2: I also suggest that the conclusions section should also include real major conclusions. It is only focused on future perspectives for research.
Response: We thank-you for pointing out that our conclusion mainly focused on future perspectives for research. To address your concern, we are now beginning the conclusion section with what is below.
Based on the results of this research, the 11-item KPPS can be used to gather data from family members to assess their initial knowledge of predatory practices related to SUD treatment, as well as their knowledge after they have completed an educational program aiming to increase their knowledge about predatory practices. This measure can also be used to assess the knowledge of treatment providers, so that interventions can be developed to enhance their knowledge about these practices, then providers can offer better services to clients. The KPPS can also be used as a screening tool to assess individuals’ knowledge about predatory practices before a referral to treatment is given for them or their loved one who is misusing substances, or a tool to assess persons’ knowledge of predatory practices who are not seeking treatment for themselves or their loved one. Further, the results of studies using this measure can be used to inform policymakers about the lack of knowledge about predatory practices among their constituents, hence having policymakers focus on developing policies that prohibit predatory practices.
Reviewer 2 Report
A better description of what the Florida legislature reference is and introduction of legislative and agency initiatives is not a complete reference more description is needed. More references of the psychobabble hotel like nature . Discuss high cost of touchy-feely rehab without adequate assessment or treatment , especially psychopharmacology . A lot of the documentation references is legislation not properly explained and generally controlled by an insurance industry that hasn't put enough thought into effective treatment but instead has a cost control lobby . agencies like NAATP are not dealing with the most fundamental issue that neuropsychological and pharmacological interventions are required in virtually all patients with a more comprehensive total health care with evidence basis! The primary test involved is probably not a validated and reliable instrument KPPS and certainly a caveat as us discussing is psychometric reliability and validity is necessary. why is the 90% female demographic so different than the demographic of normal hospital addiction? Normally there's a greater prevalence of men why is this study ~ 90% female ? A caveat section of the limitations of the study needs to be developed based on above!
The statement regarding overpromising needs to be evaluate in the context of placebo value of high rate of confidence of the delivers of care! The paper is significantly lacking data regarding the results of all these different models. 11 item Validity with .81 reliability needs better explanation from my perspective of instruments that require 175 questions to be reliable I am doubtful regardless of any statistical mumbo-jumbo! There's almost no real outcome data outcome the danger been since most patients relapse why spend any money at all. I don't believe we know yet the degree by which we can transform the lives of patients if they get brain-based total healthcare but we must try and the solution is a brain check that standardize with medical history's treatments and follow up going to the national Institute of health for a full evaluation of outcomes and results basically patients have to be filed for a lifetime to get a good outcome data. As of now the study appears week because it is missing a table of actual money spent and versus results in another table of diagnosis and treatment results when going out of town versus staying in town and other match type evaluations . I don't think the criteria for substance use disorder is adequately described and validated with a urine drug screen or some other instrument. This is essentially insurance paper of cost control without solutions board much data. Cost effectiveness is my goal but the paper does not speak to this issue and I believe must speak to the issue. And Louis Brandeis said that state governments were laboratories of experiment for the federal government to watch. entrepreneurial physicians/ rehab groups are also experiments to see what works best with this difficult patient population, which is rarely ever properly tested by either psychiatric imaging or neurocognitive a evaluations. I think more work needs to be done . And virtually all the areas described aboveAuthor Response
At the outset, we would like to say that our study did not focus on evaluating the effectiveness, cost-effectiveness, or cost-benefits of any substance abuse treatments. We were mainly interested in examining the factor structure of a new-developed measure to assess individuals’ knowledge of predatory practices, as well as the reliability of this measure. Because the focus of our paper is not on evaluating the effectiveness of any substance abuse treatment, we did not include a discussion of any treatments in our literature review.
The National Association of Addiction Treatment Providers values are as follows:
What We Value
We value a comprehensive model of care that addresses the medical, bio-psycho-social, and spiritual needs of individuals and families impacted by the disease of addiction.
We value the history of significant contributions made by 12-step abstinence-based treatment to the sobriety of over twenty million Americans in recovery.
We value residential treatment’s vital, necessary and essential place in the full continuum of care as a viable choice for the treatment of the disease of addiction.
We value a comprehensive model of care that addresses the medical, bio-psycho-social, and spiritual needs of individuals and families impacted by the disease of addiction.
We value research-driven, evidence-based treatment interventions that integrate the sciences of medicine, therapy, and spirituality. We celebrate these examples:
- Pharmaceutical interventions including medications for reducing craving and withdrawal symptoms
- Psycho-social interventions including cognitive behavioral therapy and motivational interviewing
- Spiritual interventions including Twelve Step facilitated therapy and mindfulness meditation
- Behavioral interventions including nutrition and exercise
We value abstinence from all abusable drugs as an optimal component of wellness and lifelong recovery. Depending on bio-psycho-social and economic factors, there may be persons who might require medication-assisted treatment for extended periods of time and perhaps indefinitely. However, medication alone is never sufficient to maintain long-term recovery.
We value outcome data that assess the efficacy of treatment interventions.
We value education and training that promotes understanding of a continuum of care that embraces these values.
Based on NAATP’s values, one can see that they value all types of treatment for individuals seeking treatment for substance misuse.
It should be noted that our sample did not include individuals who were in treatment. The sample consisted of their loved ones. Given that we did not assess the knowledge of predatory practices of individuals in treatment, this explains why there is a difference between our sample being predominately female as opposed to being predominately male.
Again, we would like to reiterate that purpose of this study was to examine the factor structure of the KPPS; a newly developed measure designed to assess knowledge of predatory practices related to SUD treatment. We also assessed the reliability of the measures using Cronbach Alpha. We acknowledged in the limitation section of our manuscript that we did not assess the validity of the measure and made recommendations about the types of validity to be examined, see the limitation and conclusion sections.
We have a measure that has 11 items and we had 173 respondents. Therefore, computing a Cronbach’s alpha is appropriate. We based our work on Mohamad Adam B, Evi Diana O, Nur Akmal B. A review on sample size determination for Cronbach’s alpha test: a simple guide for researchers. Malays J Med Sci. 2018;25(6):85–99. https://doi.org/10. 21315/mjms2018.25.6.9
Reviewer 3 Report
This is a very important manuscript however, it requires sloght modification:
- Invalid way of referencing according to IJERPH guidelines for authors, let me quote:
„In the text, reference numbers should be placed in square brackets [ ], and placed before the punctuation; for example [1], [1–3] or [1,3]. For embedded citations in the text with pagination, use both parentheses and brackets to indicate the reference number and page numbers; for example [5] (p. 10). or [6] (pp. 101–105).”
Please make sure you apply everything from your guidelines to your manuscript.
- The results are way too long and much of the text should be placed in discussion. Please revise the results to keep them as brief as possible.
- Please place the most important results in the abstract.
Author Response
Comment 1: Invalid way of referencing according to IJERPH guidelines for authors, let me quote:
Response: Thanks for pointing out that we had formatted the in-text citations incorrectly. We have now formatted the in-text citations correctly.
Comment 2: The results are way too long and much of the text should be placed in discussion. Please revise the results to keep them as brief as possible.
Response: We have shortened the results. Please note we have reviewed the article by Oh et al (2021) titled “The Moyamoya Health Behavior Scale for adolescent patients: Measurement tool development and psychiatric evaluation. The authors of this article examined the psychometrics of a measure, which was the purpose of our study.
Comment 3: Please place the most important results in the abstract.
Response: Based on your recommendation, we have now revised the abstract, see below.
Abstract: Given the increase in predatory practices in the substance use disorder treatment industry, calls for the development of measures to assess individuals’ knowledge about these practices. Methods: This study describes the development of the Knowledge of Predatory Practices Scale (KPPS); a newly developed measure designed to assess knowledge of predatory practices within the substance use disorder treatment industry. An exploratory factor analysis was conducted to determine the factor structure of this measure. Results: The final 11-item KPPS consists of two factors---knowledge about general predatory practices (9 items) and knowledge about unethical practices (2 items). Overall, these factors explained 67.76% of the total variance. The Cronbach’s alpha for the KPPS is 0.81. Conclusions: The KPPS is a reliable measure of knowledge of predatory practices within the substance use disorder treatment industry and can be used as a measurement tool to educate individuals seeking help for their loved ones who are misusing substances.
Round 2
Reviewer 3 Report
All of my comments were addressed.